# Systems genetics identifies a role for *Cacna2d1* regulation in elevated intraocular pressure and glaucoma susceptibility

Sumana R. Chintalapudi[1], Doaa Maria[1,2,3], Xiang Di Wang[1], Jessica N. Cooke Bailey [4],
NEIGHBORHOOD consortium, International Glaucoma Genetics consortium, Pirro G. Hysi[5], Janey L. Wiggs[6],
Robert W. Williams[1,7,8] & Monica M. Jablonski [1,2,7]

Glaucoma is a multi-factorial blinding disease in which genetic factors play an important role. Elevated intraocular pressure is a highly heritable risk factor for primary open angle glaucoma and currently the only target for glaucoma therapy. Our study helps to better understand underlying genetic and molecular mechanisms that regulate intraocular pressure, and identifies a new candidate gene, *Cacna2d1*, that modulates intraocular pressure and a promising therapeutic, pregabalin, which binds to CACNA2D1 protein and lowers intraocular pressure significantly. Because our study utilizes a genetically diverse population of mice with known sequence variants, we are able to determine that the intraocular pressure-lowering effect of pregabalin is dependent on the *Cacna2d1* haplotype. Using human genome-wide association study (GWAS) data, evidence for association of a *CACNA2D1* single-nucleotide polymorphism and primary open angle glaucoma is found. Importantly, these results demonstrate that our systems genetics approach represents an efficient method to identify genetic variation that can guide the selection of therapeutic targets.

[1] Department of Ophthalmology, The Hamilton Eye Institute, The University of Tennessee Health Science Center, Memphis, TN 38163, USA. [2] Department of Pharmaceutical Sciences, The University of Tennessee Health Science Center, Memphis, TN 38163, USA. [3] Department of Pharmaceutics, College of Pharmacy, Mansoura University, Mansoura 35516, Egypt. [4] Department of Epidemiology and Biostatistics, Institute of Computational Biology, Case Western Reserve University School of Medicine, Cleveland, OH 44106, USA. [5] Department of Twin Research and Genetic Epidemiology, King's College London, London WC2R 2LS, UK. [6] Department of Ophthalmology, Harvard Medical School, Massachusetts Eye and Ear Infirmary, Boston, MA 02298, USA. [7] Department of Anatomy and Neurobiology, The University of Tennessee Health Science Center, Memphis, TN 38163, USA. [8] Department of Genetics, Genomics and Informatics, The University of Tennessee Health Science Center, Memphis, TN 38163, USA. Correspondence and requests for materials should be addressed to M.M.J. (email: mjablonski@uthsc.edu)

Primary open angle glaucoma (POAG) is a leading cause of blindness worldwide[1]. The disease is characterized by progressive optic nerve damage arising from apoptotic cell death of retinal ganglion cells. Elevated intraocular pressure (IOP) is one of the most significant risk factors contributing to visual field loss in this disease. Steady-state IOP is generated by the balance of aqueous humor (AH) production by the ciliary body (CB) and AH drainage through the trabecular meshwork (TM; conventional pathway), and to a lesser degree the uveoscleral or nonconventional pathway. An imbalance between the inflow and outflow of AH leads to a change in IOP[2–4]. Gene variants influence an individual's likelihood of developing glaucoma, the rate of disease progression, and how a patient responds to treatment. Because IOP can be medically controlled, IOP reduction is the first-line therapeutic option in glaucoma. Current medications do not address the underlying pathologies that lead to elevated IOP, nor do they address the many potential sources of variation related to IOP modulation.

Both POAG and IOP are highly heritable. In humans, IOP heritability is estimated to be ~55%. Moreover, the genetic risk of elevated IOP and POAG are partially shared[5, 6], although some loci that are associated with POAG were not associated with IOP[7]. To date, multiple candidate IOP or POAG loci have been identified (e.g., *TMCO1*, *CDKN2B-AS1*, *GAS7*, *CAV1/CAV2*, *SIX1/SIX6*, *TXNRD2*, *ATXN2*, *FOXC1 ABCA1*, *AFAP1*, *GMDS*, *PMM2*, *TGFBR3 ARHGEF12*, *FAM125B*, *FNDC3B*, and the *ABO* blood group), however, their physiological roles are not well understood[7–16]. Identification of additional gene variants that modulate IOP both in animals and humans is therefore likely to provide critical insights and new targets for therapeutic intervention.

Systems genetics is an extension of complex trait analysis that examines large sets of genotypes and phenotypes to investigate the genetic basis of disease traits[17–20]. The BXD family is currently the largest and best characterized mouse genetic reference population[21]. This family is an admixture of C57BL/6J and DBA/2J genomes, and the family are segregating for roughly 5.5 million sequence variants[22]. Over the last decade, the BXD family has become a key resource for systems genetics largely because they have been phenotyped so thoroughly and at many levels—from messenger RNA (mRNA) to maternal behavior[17, 23–25]. There is also extensive phenome data on the visual system of these strains[26–30]. One of the major advantages of the BXDs relative to the Collaborative Cross and other new resources is that the DBA/2J progenitor strain and many of the BXD progeny strains consistently develop high IOP between 6 and 10 months of age[31]. Therefore, they are an ideal resource to discover gene variants that modulate both IOP and glaucoma.

In this study, we systematically measure IOP across a large subset of the BXD family in multiple age cohorts. Using stringent stepwise refinement based on expression quantitative trait locus (QTL) mapping, correlation analyses (direct and partial Pearson test), and the analysis of single-nucleotide polymorphisms (SNPs), we identify a candidate gene that modulates IOP, and we combine mouse and human genetic data in an effort to validate the candidate gene. Lastly, we evaluate the IOP-lowering effect of a drug specifically targeted to the candidate protein product. Collectively, our study finds that *Cacna2d1* modulates IOP and that blocking the function of its gene product, CACNA2D1, with pregabalin reduces IOP in a dose-dependent and haplotype-specific manner.

## Results

**Study design**. A major advantage of the genetically diverse BXD family over non-inbred strains is that each strain is homozygous

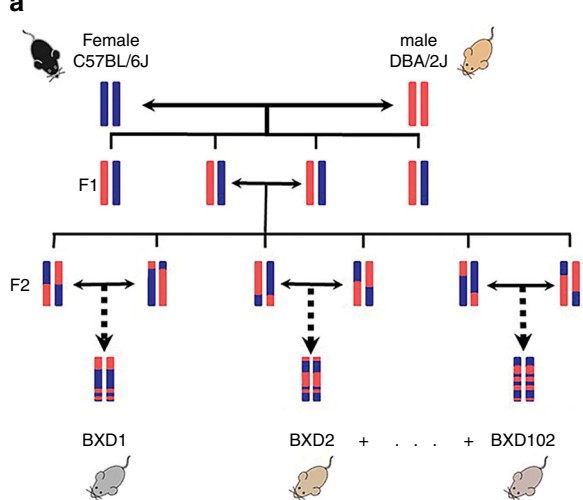

**a**

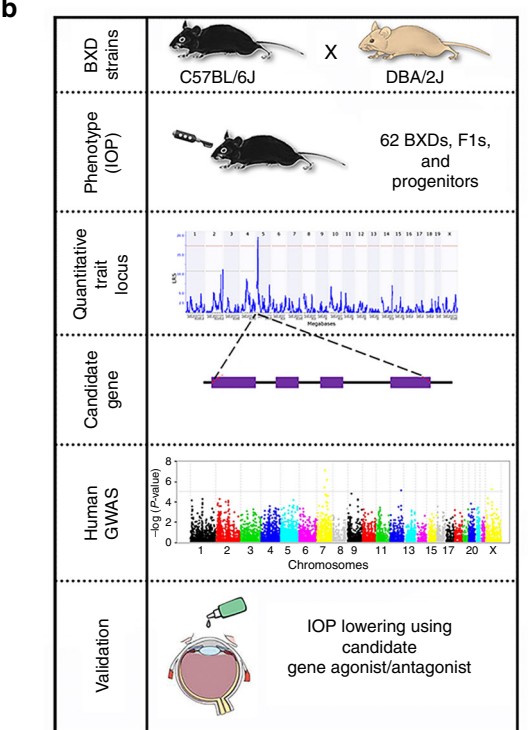

**b**

**Fig. 1** Overview of IOP systems genetics analyses. **a** BXD strains derived by crossing C57BL/6J (B6) and DBA/2J (D2) parents. The resulting heterozygous F1 mice were crossed to generate genetically diverse but non-reproducible F2 animals. These F2 progeny were iteratively inbred until generation F20+, at which point each strain represents a unique mosaic of *B* and *D* alleles. **b** Workflow for IOP systems genetics analysis. IOP was measured from 65 BXD strains aged 9.1–13 months. Genomic regions modulating IOP were identified using QTL analyses. Stringent refinement based on QTL mapping, correlation analyses, and SNPs was performed to identify positional candidate genes. Subcellular localization of candidates in mouse and human eyes were determined by immunohistochemistry. The NEIGHBORHOOD consortium database was used to identify SNPs within the candidate gene associated with elevated IOP and POAG in humans. The IOP-lowering effect of pregabalin, a drug with high affinity for CACNA2D1 was evaluated as eye drops in D2, BXD48 (*D* allele), B6, and BXD14 (*B* allele) strains for validation

across its genome (Fig. 1a), which allows for an infinite source of genetically identical test subjects. IOP is one factor that can influence glaucoma risk. To identify a gene candidate that modulates IOP, we utilized a systems genetics, pharmacological, and translational approach (Fig. 1b). We have successfully used this approach in previous studies to identify gene modulators of various traits in BXD mice[26, 27, 29, 30, 32, 33].

**Trait and genetic variation across BXD mice.** IOP varied over twofold in 9.1–13-month-old BXD strains (range of $9.6 \pm 0.92$

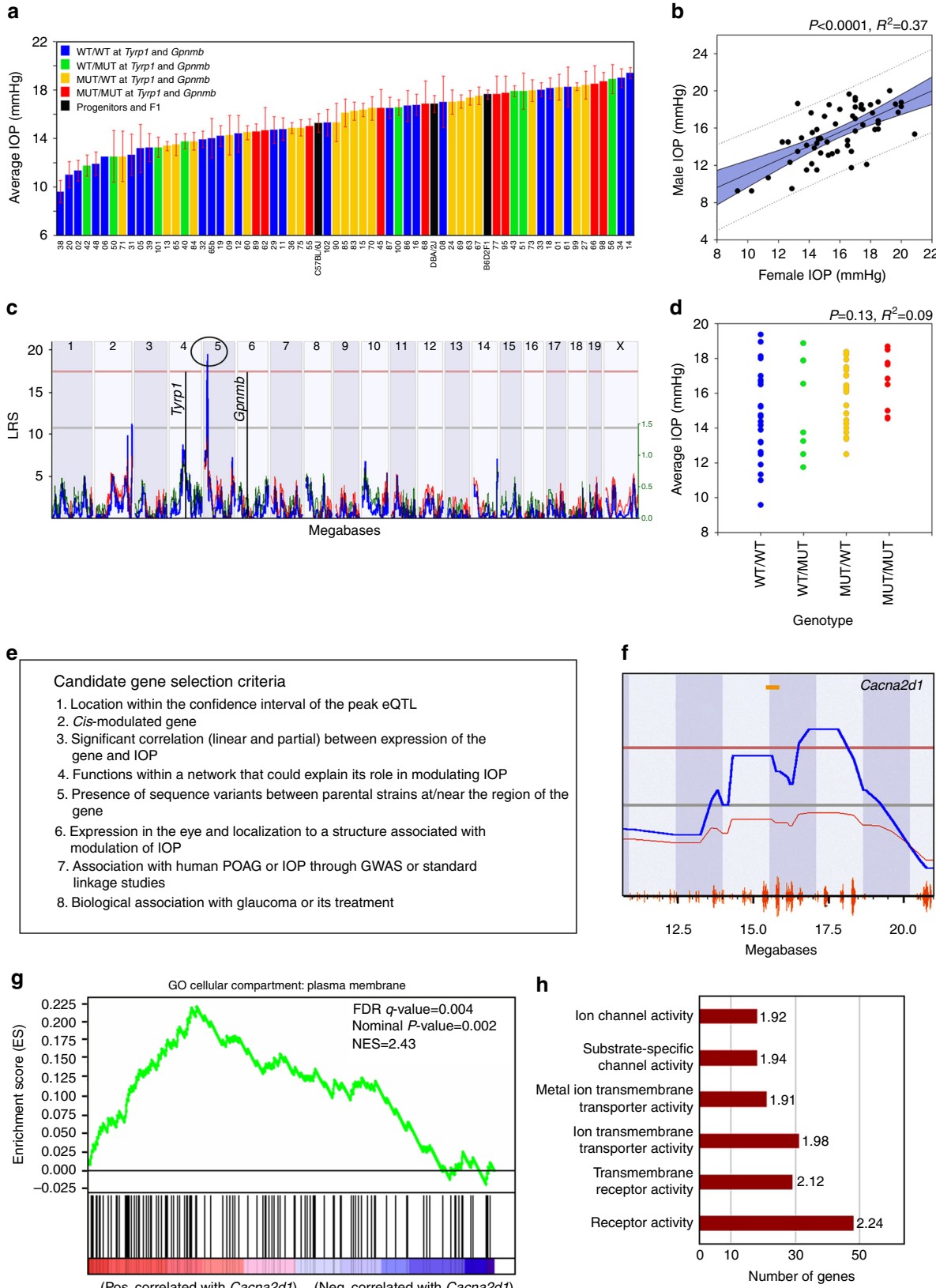

mm Hg in BXD38 to $19.38 \pm 0.43$ mm Hg in BXD14; Fig. 2a). IOP heritability was 31%, which is comparable to that of other ocular traits to which QTL mapping has been applied successfully[27, 29, 32–34]. There was no statistically significant difference in IOP between the sexes (Fig. 2b).

The variation in IOP was mapped to a narrow locus on proximal Chr 5 (likelihood ratio statistic (LRS) = 19.6, logarithm of the odds (LOD) = 4.25, Fig. 2c). This QTL was not associated with IOP in studies using younger sex-matched BXD strains, suggesting that the QTL contains gene variants that contribute to variation in IOP at an older age[29]. In addition, the location of this LRS interval does not overlap with the location of either *Tyrp1* or *Gpnmb* (Fig. 2c). As confirmation that the well-documented mutations in *Tyrp1* and *Gpnmb* do not influence IOP in the BXD GRP, IOP also maps to the same location on Chr 5 after we exclude strains that harbor both mutations or harbor only one of the mutations (Supplementary Fig. 1). Moreover, the complete overlap of IOP distributions for all possible genotypes at those two loci ($P = 0.13$; Fig. 2d) further supports this claim.

**Selection of positional candidates in the chromosome 5 locus.** To determine the candidate gene variants that modulate IOP within the Chr 5 locus, we used the following stringent criteria (Fig. 2e): (1) the gene is located within the confidence interval of the peak eQTL; (2) the gene has *cis*-modulation; (3) the expression level of the gene across BXD strains is significantly correlated with elevated IOP using both linear correlation and partial Pearson correlation analyses; (4) the gene functions within a network that could explain its role in modulating IOP; (5) the gene has sequence variants between parental strains at/near the region of the gene; (6) the gene is expressed in the eye and localized to an area associated with modulation of IOP; (7) the gene is associated with human POAG and/or elevated IOP either through GWAS or standard linkage studies; and (8) the gene has a biological association with glaucoma or its treatment.

Within the QTL peak at Chr 5, there were 25 positional gene candidates that were *cis*-regulated (Supplementary Table 1). Using our above criteria, calcium channel, voltage-dependent, α2δ1subunit (*Cacna2d1*) emerged as the single best positional candidate ($r = 0.440$; $P = 0.0003$; Fig. 2f). No other positional candidate fulfilled our selection criteria (Supplementary Table 2). Gene set enrichment analyses (GSEA; Broad Institute[35]) determined that nominally significant gene correlates of *Cacna2d1* (Supplementary Data 1) were significantly enriched for plasma membrane localization (Fig. 2g) and for functions related to neuronal function and excitability, including ion channel activity, substrate-specific channel activity, metal ion transmembrane transporter activity, ion transmembrane transporter activity, and receptor activity (Fig. 2h). These results suggest that

*Cacna2d1* likely functions within pathways that critically influence ion channels and/or their receptors within the eye that affect IOP regulation.

**Gene haplotypes of *Cacna2d1* contribute to variations in IOP.** The expression of the *Cacna2d1* transcript in the whole eye varied significantly among BXD strains (range of $6.9 \pm 0.2$ in BXD71 to $9.7 \pm 0.3$ in BXD48a), which is a 5.6-fold difference in mRNA expression (Fig. 3a). BXD strains with the *D* parental allele have lower expression of *Cacna2d1* in the eye than those strains with the *B* parental allele (Fig. 3a, b, $P = 1.2 \times 10^{-32}$). *Cacna2d1* mapped as a significant *cis*-eQTL (LRS = 143; LOD = 31.01) on proximal Chr 5 at 14.3 Mb, which is within 5 Mb of location of the gene itself (Fig. 3c), making it a *cis*-regulated gene.

*Cacna2d1* encodes for a preproprotein that is cleaved into multiple chains that form the alpha-2 and delta subunits of the voltage-dependent calcium channel complex. CACNA2D1 is a glycosylphosphatidylinositol (GPI)-anchored subunit typically associated with the Cavα1 pore within L-type calcium channels in smooth muscle (Fig. 3d). The gene is highly polymorphic (Fig. 3e) with 1056 SNPs and 30 insertions/deletions between the parent strains (Supplementary Data 2). These variants segregate among the BXD strains based upon the haplotype of the gene. Similarly, in humans, *CACNA2D1* is highly polymorphic and has several splice variants[36–38].

We further sought to determine if the *Cacna2d1* haplotype influenced the baseline IOP in BXD mice. Similar to the influence of the parental allele on *Cacna2d1* expression (Fig. 3b), we found a distinct segregation of IOP values among the BXD strains that was dependent upon the haplotype of *Cacna2d1* that was age-dependent. The *Cacna2d1* haplotype had no significant effect on IOP in younger BXD strains ($P = 0.34$, age 1–2.1 months) (Fig. 3f, left). In contrast, in older age (aged 9.1–13 months), strains with *Cacna2d1* inherited from the B6 parent (*B* allele) had significantly higher IOP than those with the *D* allele ($P = 0.0008$, Fig. 3f, right).

To further assess the strength of this candidate gene, we evaluated human POAG GWAS data from the NEIGHBOR-HOOD consortium ($N = 3853$ cases and 33 480 controls)[16]. A total of 1520 SNPs in the human genomic region that includes *CACNA2D1* (chromosome 7:81.9–82.4 Mb) were evaluated, identifying nominal association for POAG ($P < 0.05$) for 44 SNPs (Supplementary Table 3), with the lead SNP (rs2299184 [A], $P = 0.001$, odds ratio = 1.15) located in intron 1 near DNaseI hypersensitivity sites annotated by the Encyclopedia of DNA Elements (ENCODE) as active in non-pigmented ciliary epithelium. Eight of the 44 human *CACNA2D1* SNPs showing nominal association in the NEIGHBORHOOD POAG GWAS were also included in a multi-ethnic IOP association study[9], however,

**Fig. 2** Association analyses reveal *Cacna2d1* as a candidate for IOP modulation. **a** IOP levels vary among the BXD strains. IOP of BXD strains carrying wild-type alleles of *Tyrp1* and *Gpnmb* (WT/WT; blue bars), wild-type *Tyrp1* and mutant *Gpnmb* (WT/MUT; green bars), mutant *Tyrp1* and WT *Gpnmb* (MUT/WT; yellow bars), and mutant alleles of both genes (MUT/MUT; red bars) are shown. $n = 65$ strains, age = 9.1–13 months. Values denote IOP levels on mmHg scale (mean ± SEM). **b** No differences in IOP between sexes. There is a strong statistical association between the IOP of males and females ($P < 0.0001$). The relationship between the sexes is positively correlated ($R^2 = 0.37$, Pearson correlation coefficient = 0.62). **c** Genetic interval mapping revealed a single significant eQTL on proximal Chr 5. This is distinct from the locations of *Tyrp1* and *Gpnmb* in the genome (black vertical lines on Chr 4 and Chr 6, respectively). **d** *Tyrp1* and *Gpnmb* haplotypes do not influence IOP in BXD mice. The scatter plot shows average IOP of BXD strains carrying wild-type alleles of *Tyrp1* and *Gpnmb* (WT/WT), wild-type allele of *Tyrp1* and mutant allele of *Gpnmb* (MUT/WT), mutant allele of *Tyrp1* and wild-type allele of *Gpnmb* (WT/MUT), and mutant alleles of both genes (MUT/MUT) ($n = 65$, $P = 0.13$, age = 9.1–13 months). *P*-value was calculated using an ANOVA. **e** Stringent selection criteria for selecting candidate genes. **f** A significant QTL for IOP is present on chromosome 5 between 14–19 Mb. *Cacna2d1* is the strongest candidate gene located in the peak QTL. **g** Gene set enrichment analysis for positive correlates of *Cacna2d1*. Gene correlates of *Cacna2d1* were significantly enriched for localization to the plasma membrane. False discovery rate (FDR) cutoff was set as $q \leq 0.25$. **h** Gene set enrichment analysis of genes positively correlated with *Cacna2d1* presented as molecular function groupings. Categories that are statistically over-represented are shown with their normalized enrichment score (NES) listed next to the bars

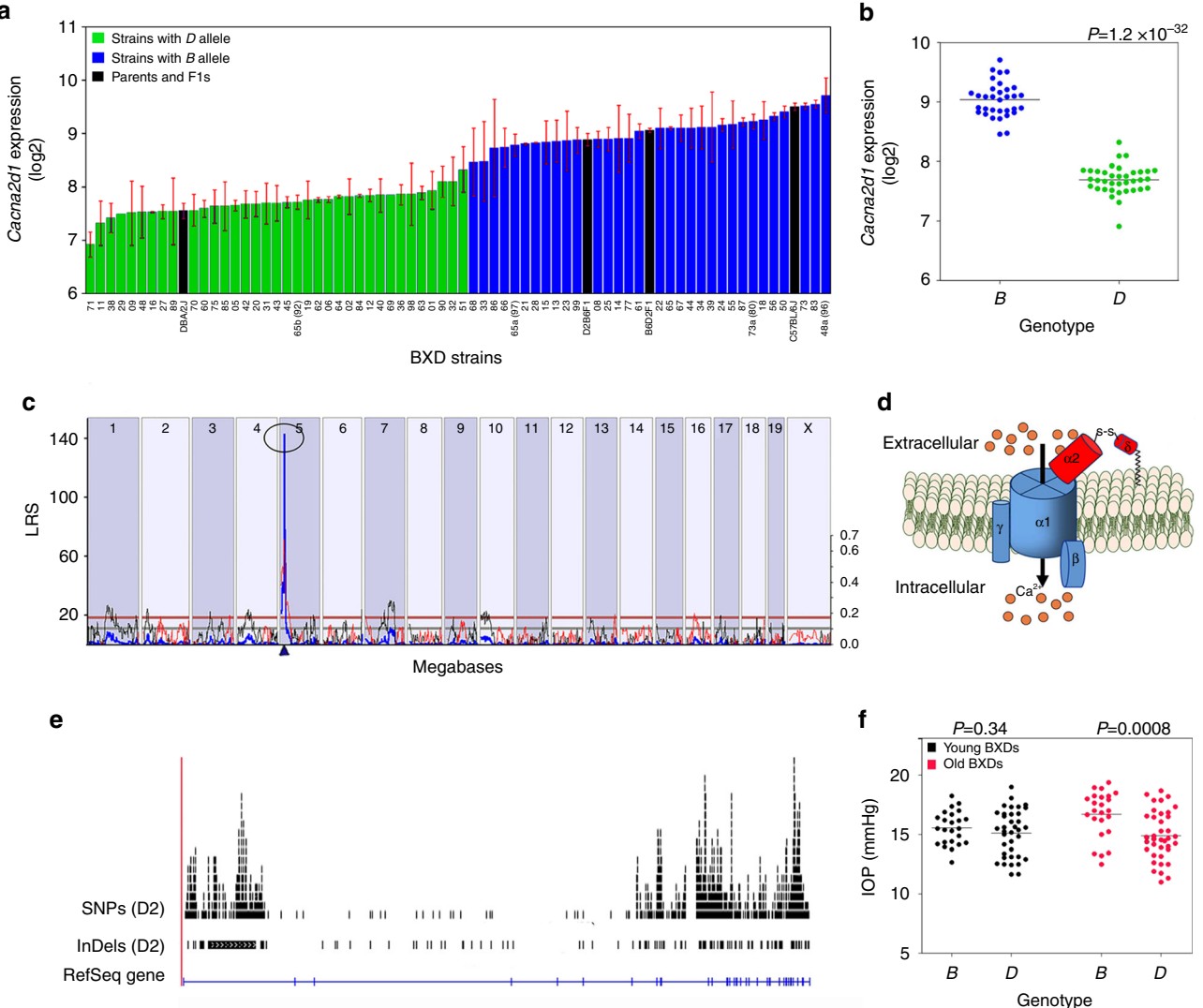

**Fig. 3** Association analyses of *Cacna2d1* haplotype variants. **a** *Cacna2d1* expression in the whole eye varies across BXD strains. The bars depict a range of expression values from $6.9 \pm 0.03$ and $9.7 \pm 0.10$ (mean±SEM). On the Y-axis, *Cacna2d1* expression is on a $\log_2$ scale. Parental strains (black), F1s (black), *D* parental allele *Cacna2d1* (green), and *B* parental allele of *Cacna2d1* (red) are on X-axis. **b** *Cacna2d1* haplotype influences *Cacna2d1* expression level in eyes of BXD strains. BXD strains with the *D* parental allele have lower expression of *Cacna2d1* in the eye, while those with the *B* parental allele have higher expression of the gene ($n = 70$; $P < 0.0001$). *P*-value was calculated using an ANOVA. **c** Genetic mapping revealed a single highly significant *cis*-eQTL for *Cacna2d1* on Chr 5. The purple triangle indicates the location of *Cacna2d1* within the mouse genome. **d** Subunit assembly of voltage-gated calcium channels. Graphic representation of the high voltage-activated calcium channel complex consisting of the main pore forming $\alpha_1$ (blue) subunit plus ancillary, $\beta$ (blue), $\gamma$ (blue), and $\alpha_2$ (red) and $\delta_1$ (red) subunits. $\alpha_2$ and $\delta_1$ subunits have a disulphide bond between them. The $\delta_1$ unit is GPI-anchored to the plasma membrane. **e** UCSC Genome Browser illustration of gene structure, and location of SNPs and InDels in *Cacna2d1* on Chr 5 of the mouse genome. Reference Sequence mRNA is represented in blue. **f** Effect of *Cacna2d1* haplotype on IOP is age-dependent. In this scatter plot, parental allele of *Cacna2d1* does not significantly influence IOP in young BXDs ($P = 0.34$, $n = 63$, age = 1–2.1 months; left). In contrast, BXD strains carrying the *B* parental allele of *Cacna2d1* have higher IOP, while those with the *D* parental allele have lower IOP in older mice ($P = 0.0008$, $n = 64$, age = 9.1–13 months; right). *P*-value was calculated using an ANOVA. F1s were not included in these analyses

significant association for these eight SNPs was not found in the IOP metadata, possibly due to different genetic effects in the multiple ethnicities included in that study.

**CACNA2D1 is localized to the CB and TM in mice and humans**. To further test *Cacna2d1* as a candidate IOP-modulating gene, we performed immunohistochemistry to determine the localization pattern of CACNA2D1 in healthy mouse and human donor eyes. In the mouse eye, CACNA2D1 is prominently localized to the TM, CB, and ciliary muscle (CM) (Fig. 4a–c). CACNA2D1 was observed in a punctate pattern throughout the TM and Schlemm's canal. In the CB, CACNA2D1

was highly expressed in the non-pigmented epithelium. Weak labeling was present in the CM. A similar pattern of expression was observed in the TM and CB of the human donor eye (Fig. 4d–f).

**IOP response to pregabalin depends on *Cacna2d1* genotype**. Based upon our data suggesting that *Cacna2d1* modulates IOP, we evaluated the ability of pregabalin, a gabapentinoid drug with high specificity for CACNA2D1, to affect IOP. Our data demonstrate that pregabalin ophthalmic eye drops reduce IOP in mice in a dose-dependent manner (Fig. 5a). The percent reduction in IOP for both treated and control eyes after application of

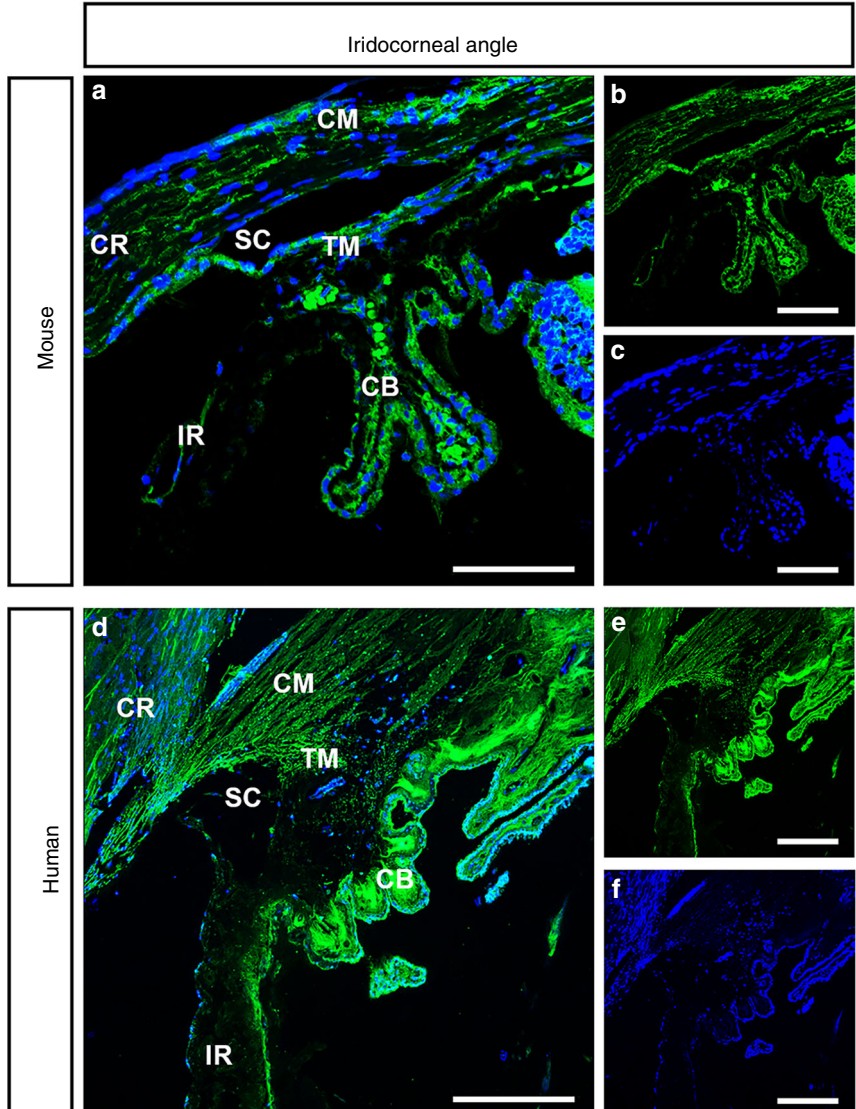

**Fig. 4** Cellular localization of CACNA2D1 in C57BL/6 J mouse and human donor eyes. Cellular localization of CACNA2D1 in C57BL/6J mouse and human donor eyes. **a–f** Sections from C57BL/6J mouse (**a–c**) and human donor (**d–f**) iridocorneal angle were labeled with anti-CACNA2D1 antibodies. CACNA2D1 (green) is localized in the ciliary body (CB), trabecular meshwork (TM), Schlemm's canal (SC), and ciliary muscle (CM) in both B6 mice and human donor eyes. CACNA2D1 was also present in the posterior pigmented epithelium of the iris. $n = 2$ mice and 1 human. Scale: 100 μm. Blue nuclei, CB ciliary body, CM ciliary muscle, CR cornea, IR iris, SC Schlemm's canal, TM trabecular meshwork

our ophthalmic pregabalin eye drops (0.3–1.2%) are shown in Fig. 5a. Table 1 lists the pharmacodynamic parameters after application of pregabalin eye drops and statistical evaluation of the data, respectively. Drops containing 0.3% drug provided no IOP-lowering effect compared to control. All other concentrations of drug reduced IOP in a dose-dependent manner. A plateau was reached at 0.9% and there was no significant difference in drug response between 0.9 and 1.2% drug ($P > 0.05$). There was no significant difference between the time of maximum response ($T_{max}$) values of 0.6–1.2% concentrations of drug ($P > 0.05$). In contrast, there was a significant difference between the time required for IOP to return to baseline ($T_{end}$) for all concentrations of pregabalin eye drops, ($P < 0.0001$). The 1.2% pregabalin eye drops extended the duration of the IOP-lowering effect of pregabalin above that obtained with 0.6% pregabalin. Because there was no significant difference in the percent reduction of IOP between 0.9 and 1.2% pregabalin, we selected 0.9% as the minimal concentration required to produce the maximum reduction in IOP.

The *Cacna2d1* haplotype also influenced the drug response. Figure 5b illustrates the percent reduction in IOP after application of 0.9% pregabalin eye drops into the eyes of DBA/2J, BXD48 (*D* allele), C57BL/6J, and BXD14 (*B* allele) mice. Table 2 lists the pharmacodynamic parameters after application of 0.9% pregabalin eye drops and statistical evaluation of the data, respectively. Mice with the *B* allele of *Cacna2d1* (i.e., B6 and BXD14) were more responsive to pregabalin (0.9%) eye drops than mice with the *D* allele (i.e., D2 and BXD48). In addition to larger reductions in IOP, mice with the *B* allele of *Cacna2d1* had larger $T_{max}$, $T_{end}$, and area under the curve ($AUC_{total}$) than mice with the *D* allele. Expanding this analysis to an additional species, we observed a similar IOP-lowering response ($22.1 \pm 2.8\%$) in Dutch belted rabbits after instillation of 0.9% pregabalin eye drops (Fig. 5c). Table 3 lists the pharmacodynamic parameters and statistical evaluation of the data after application of 0.9% pregabalin eye drops into the eyes of Dutch belted rabbits.

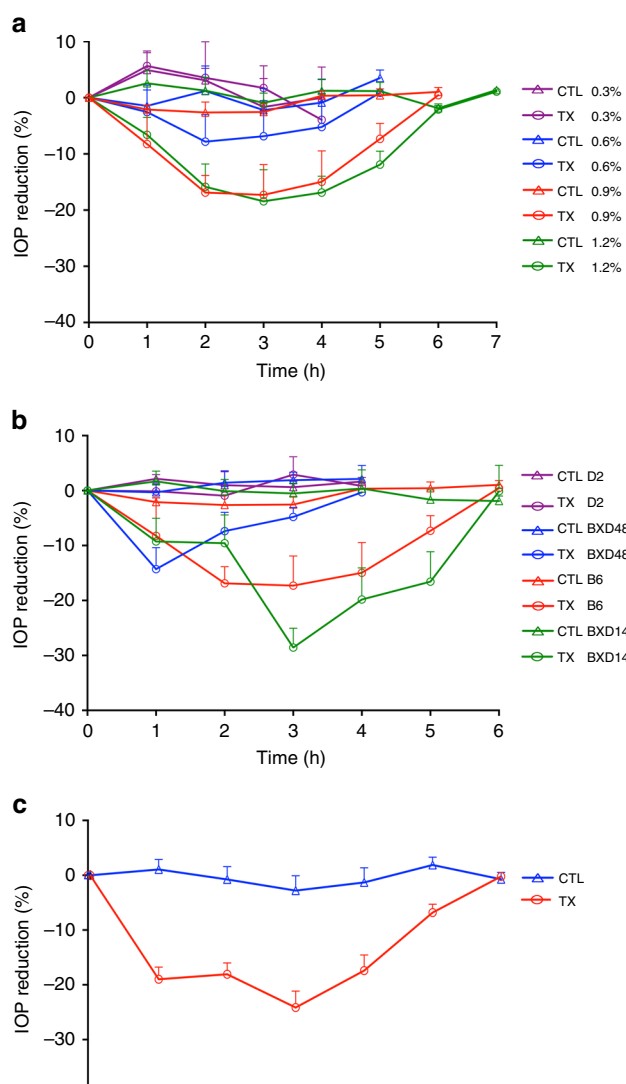

**Fig. 5** *Cacna2d1* haplotype effects IOP-lowering potency of pregabalin, a specific modulator of CACNA2D1, lowers IOP in B6 mice in a dose-dependent manner. **a** We measured a dose-dependent reduction in IOP compared to control treatment after a single application of ophthalmic formulation containing a range of concentrations of pregabalin from 0.3 to 1.2%. The minimal concentration required to produce the maximum reduction in IOP is 0.9% (mean ± SEM; $n = 6$). Statistical details are provided in Table 1. **b** Pregabalin-induced IOP reduction is haplotype-specific. Mice carrying the *B* parental allele of *Cacna2d1* (i.e., B6 and BXD14) are more responsive to pregabalin (0.9%) than mice carrying the *D* parental allele (i.e., D2 and BXD48; mean ± SEM; $n = 4$–6). Statistical details are provided in Table 2. **c** A single dose of pregabalin eye drops (0.9%) lowers IOP by 22.1% in Dutch belted rabbits. $n = 5$. Statistical details are provided in Table 3

## Discussion

By combining a forward murine genetics approach with cell biology, pharmacology, and analysis of human GWAS data, we were able to identify a genetic locus that significantly modulates IOP in the BXD murine genetic reference panel. Furthermore, we have demonstrated that our top candidate gene—*Cacna2d1*—is critical for modulation of IOP in these strains. Although other positional candidates in our Chr 5 QTL, such as *Nos3* and *Sema3e*

(Supplementary Table 1), have been linked to glaucoma[39–41], they do not fulfill our stringent selection criteria and therefore have been eliminated as candidate genes (Supplementary Table 2). Other positional candidates, such as *Crygn* and *Klhl7*, have been associated with other ocular diseases[42, 43], but not glaucoma.

We have validated *Cacna2d1* using multiple stringent criteria and identified human SNPs in the CACNA2D1 genomic region that are nominally associated with POAG. Furthermore, we have demonstrated that both baseline IOP and steady-state expression of *Cacna2d1* are dependent upon the parental allele of the gene in mice. As a corollary, we further show that the response to pregabalin, a gabapentinoid drug with high affinity for CACNA2D1, is also dependent upon the *Cacna2d1* haplotype. This is a bidirectional study that has successfully identified an IOP-modulating candidate gene in glaucoma and demonstrated a differential response to therapy. Our results further underscore the potential value of using the BXD genetic reference panel as a powerful tool to study human disease. Collectively, our data identify *CACNA2D1* as a modulator of IOP and provide an avenue for a precision medicine approach to glaucoma therapy.

Previous studies have demonstrated a link between $Ca^{2+}$, calcium channels and glaucoma[44–53], providing a historical backdrop for our discovery of the role of *Cacna2d1* in IOP modulation. Because $Ca^{2+}$ likely plays a role in the pathophysiology of glaucoma[45], systemic calcium channel blockers (CCBs) (e.g., verapamil, diltiazem, or nimodipine) have been evaluated as plausible therapies for POAG. However, the outcomes of these investigations have been inconsistent, with some studies demonstrating that CCBs are effective in lowering IOP, protecting ganglion cells, increasing ocular blood flow and improving visual function while others fail to replicate those results[46–53]. In all studies performed to date, however, CCBs have targeted the α1 pore of the $Ca^{2+}$ channel and have used a systemic route of administration. None of the previous investigations evaluated modulators of the $Ca^{2+}$ channel auxiliary subunits, such as CACNA2D1, nor have they evaluated topical delivery of the drug. In our study, we demonstrate that targeting CACNA2D1 with topical pregabalin lowers IOP up to 30% in mice with the *B* haplotype of the gene, which is a significantly higher effect than that demonstrated using systemically administered traditional CCBs.

Previous human genomic studies (GWAS and linkage studies) revealed multiple genomic regions that associated with elevated IOP and/or POAG[7, 9–11, 16]. Two regions of interest lie on Chr 7q near the locations of *CACNA2D1*[54–56] and *CAV1/CAV2*[7]. Other studies have demonstrated that SNPs in *Cacna2d1/CACNA2D1* are associated with short QT syndrome[38], perifosine cytotoxicity[57], altered sensitivity to acute noxious heat[58], and beef quality traits[59], indicating that polymorphisms in this gene have direct and distinct physiological consequences, some of which may be associated with modulation of IOP.

We show that CACNA2D1 is expressed in both CB and TM, the sites of AH production, and outflow through the traditional pathway, respectively, thus suggesting a possible role in AH dynamics. It is also modestly present in the CM, a structure that directly affects the distention of the TM in an antagonistic manner[3]. We also demonstrate that the *Cacna2d1* likely functions in a network that includes other modulators of metal ion channel and transport activity. Based upon these collective data, we propose a model for the role of CACNA2D1 in regulating IOP (Fig. 6). Pregabalin, or other gabapentinoid drugs, binds to the CACNA2D1 subunit of the calcium channel, the affinity of which varies depending on the haplotype of the gene. The binding of the drug mitigates the flux of $Ca^{2+}$ through the α1 pore of the calcium channel, reducing the level of intracellular $Ca^{2+}$. A reduction in $Ca^{2+}$ could cause a concomitant reduction in ion transport, water

**Table 1 Pharmacodynamic parameters after application of drops containing four concentrations of pregabalin to the eyes of B6 mice**

| Pharmacodynamic parameters (mean ± SEM) | Pregabalin eye drops | | | |
|---|---|---|---|---|
| | 0.3% | 0.6% | 0.9% | 1.2% |
| Baseline IOP (mmHg) | 15.4 ± 0.3 | 15.4 ± 0.1 | 15.3 ± 0.1 | 15.5 ± 0.1 |
| IOP at $T_{max}$ (mmHg) | — | 14.2 ± 0.7 | 12.7 ± 0.8 | 12.6 ± 0.8 |
| ΔIOP (mmHg) | — | −1.2 ± 0.8 | −2.7 ± 0.8 | −2.9 ± 0.9 |
| % Reduction in IOP | — | 7.8 ± 4.8 | 17.3 ± 5.4 | 18.4 ± 5.6 |
| $T_{max}$ (h) | — | 3.0 ± 0.4 | 2.8 ± 0.5 | 3.4 ± 0.4 |
| $T_{end}$ (h) | — | 5.5 ± 0.3 | 6.0 ± 0.4 | 6.8 ± 0.2 |
| $AUC_{total}$ (% h) | — | 42.9 ± 11.1 | 67.1 ± 19.1 | 71.9 ± 12.1 |

**Statistical comparisons among four concentrations of pregabalin drops applied to the eyes of B6 mice**

| Pharmacodynamic parameters (mean ± SEM) | Overall P-value | 0.3% vs. 0.6% | 0.3% vs. 0.9% | 0.3% vs. 1.2% | 0.6% vs. 0.9% | 0.6% vs. 1.2% | 0.9% vs. 1.2% |
|---|---|---|---|---|---|---|---|
| % Reduction in IOP | 0.034 | >0.05 | >0.05 | <0.05 | >0.05 | >0.05 | >0.05 |
| $T_{max}$ (h) | <0.0001 | <0.001 | <0.001 | <0.0001 | >0.05 | >0.05 | >0.05 |
| $T_{end}$ (h) | <0.0001 | <0.0001 | <0.0001 | <0.0001 | >0.05 | <0.05 | >0.05 |
| $AUC_{total}$ (% h) | 0.002 | >0.05 | <0.01 | <0.01 | >0.05 | >0.05 | >0.05 |

Overall P-value represents the outcome of the one-way ANOVA analysis. Individual P-values represent the outcome of Tukey–Kramer multiple comparisons tests. $AUC_{total}$ (% h) area under the curve in percent IOP reduction x hours, $T_{end}$ (h) time to end of response in hours, $T_{max}$ (h) time to maximum response in hours

**Table 2 Pharmacodynamic parameters after application of 0.9% pregabalin eye drops to D2, BXD48 (D allele), B6, and BXD14 (B allele) mice**

| Pharmacodynamic parameters (mean ± SEM) | Mouse strain | | | |
|---|---|---|---|---|
| | D2 | BXD48 | B6 | BXD14 |
| Baseline IOP (mmHg) | 17.2 ± 0.3 | 16.8 ± 0.5 | 15.3 ± 0.1 | 17.8 ± 0.6 |
| IOP at $T_{max}$ (mmHg) | — | 14.4 ± 0.7 | 12.7 ± 0.8 | 12.7 ± 0.5 |
| ΔIOP (mmHg) | — | −2.4 ± 0.6 | −2.7 ± 0.8 | −5.1 ± 0.8 |
| % Reduction in IOP | — | 14.3 ± 3.9 | 17.3 ± 5.4 | 28.6 ± 3.5 |
| $T_{max}$ (h) | — | 1.5 ± 0.2 | 2.8 ± 0.5 | 3.2 ± 0.3 |
| $T_{end}$ (h) | — | 3.0 ± 0.5 | 6.0 ± 0.4 | 6.5 ± 0.3 |
| $AUC_{total}$ (% h) | — | 26.6 ± 4.5 | 67.1 ± 19.1 | 88.7 ± 12.6 |

**Statistical comparisons among different strains of mice after application of 0.9% pregabalin eye drops**

| Pharmacodynamic parameters (mean ± SEM) | Overall P-value | D2 vs. BXD48 | D2 vs. B6 | D2 vs. BXD14 | BXD48 vs. B6 | BXD48 vs. BXD14 | B6 vs. BXD14 |
|---|---|---|---|---|---|---|---|
| % Reduction in IOP | 0.001 | >0.05 | >0.05 | <0.001 | >0.05 | <0.05 | >0.05 |
| $T_{max}$ (h) | <0.0001 | <0.05 | <0.001 | <0.0001 | <0.05 | <0.01 | >0.05 |
| $T_{end}$ (h) | <0.0001 | <0.001 | <0.0001 | <0.0001 | <0.001 | <0.0001 | >0.05 |
| $AUC_{total}$ (% h) | 0.0002 | >0.05 | <0.01 | <0.001 | >0.05 | <0.01 | >0.05 |

Overall P-value represents the outcome of the one-way ANOVA analysis. Individual AUC-values represent the outcome of Tukey–Kramer multiple comparisons tests. $AUC_{total}$ (% h) area under the curve in percent IOP reduction x hours, $T_{end}$ (h) time to end of response in hours, $T_{max}$ (h) time to maximum response in hours

**Table 3 Pharmacodynamic parameters after application of 0.9% pregabalin eye drops to Dutch belted rabbits**

| Pharmacodynamic parameters | (Mean ± SEM) |
|---|---|
| Baseline IOP (mmHg) | 20.7 ± 0.5 |
| IOP at $T_{max}$ (mmHg) | 16.1 ± 1.4 |
| ΔIOP (mmHg) | −4.6 ± 0.6 |
| % Reduction in IOP | 22.1 ± 2.8 |
| $T_{max}$ (h) | 2.4 ± 0.6 |
| $T_{end}$ (h) | 6.0 ± 0.0 |
| $AUC_{total}$ (% h) | 78.2 ± 4.2 |

$AUC_{total}$ (% h) area under the curve in percent IOP reduction x hours, $T_{end}$ (h) time to end of response in hours, $T_{max}$ (h) time to maximum response in hours

transfer, and contraction of smooth muscle cells. The net result of pregabalin binding to CACNA2D1 could be threefold: a reduction in the production of AH by the CB; a relaxation of the TM and resultant facilitation of the outflow of AH; and relaxation of the CM, which would have an antagonistic effect on TM relaxation. Given the marked effect that pregabalin has on baseline IOP in BXD mice carrying the B haplotype of Cacna2d1, we predict that the balance between the effect of the drug on TM cells and also CM cells would be in favor of increased outflow through the TM. If our hypothesis is valid, pregabalin binding to both CB and TM, could both reduce AH production and increase AH outflow.

In summary, our study identified Cacna2d1 as a modifier of IOP. We offer a hypothesis regarding the modulation of IOP and

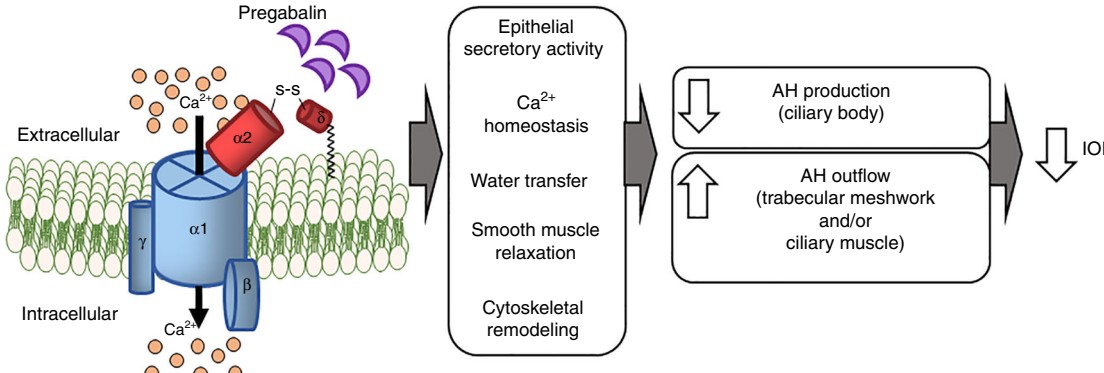

**Fig. 6** Proposed mechanism of IOP reduction by pregabalin. We predict that pregabalin binds to CACNA2D1 localized to CB, TM, and/or CM cells. Pregabalin binding causes the α1 pore to close, leading to a decrease in $Ca^{2+}$ influx into cells and a resultant decrease in free cytosolic $Ca^{2+}$. Consequently, the cells relax and aqueous humor inflow may be reduced and/or outflow may be increased, leading to a reduction in IOP

offer a potential therapeutic target that could preserve vision by reducing IOP. In the future, a similar approach could provide a bridge between systems biology and effective drug target identification.

## Methods

**Animals**. A total of 65 BXD strains, parents, and F1 crosses (548 mice at 9.1–13 months age) were used for QTL mapping of IOP in mice aged 9.1–13 months. The gender distribution was relatively equal (313 females and 235 males). IOP measured from 64 BXD strains aged 1–2.1 months (data available on GeneNetwork) were used for comparison. The parental strains C57BL/6J ($n = 6$; aged 3–5 months), DBA/2J ($n = 4$; aged 3–5 months), and two BXD strains—BXD14 and BXD48 ($n = 6$; aged 3–5 months)—were used to evaluate the IOP-lowering effects of pregabalin. C57BL/6 J and DBA/2 J were used as controls. BXD14 and BXD48 strains were selected based on presence of the *B* and *D* haplotype of *Cacna2d1*, respectively. Dutch belted rabbits ($n = 5$; procured from Robinson Services, Mocksville, NC, USA) weighing ~2.0–2.5 kg were used to test IOP-lowering effects of pregabalin. All procedures involving mice and rabbits were approved by the Animal Care and Use review board of the University of Tennessee Health Science Center (UTHSC) and followed the Association of Research in Vision and Ophthalmology Statement for the Use of Animals in Ophthalmic and Vision Research in addition to the guidelines for laboratory animal experiments (Institute of Laboratory Animal Resources, Public Health Service Policy on Humane Care and Use of Laboratory Animals). Animals were housed under cyclic light (12 h on:12 h off) with 35% humidity in a specific pathogen-free facility at UTHSC, and were allowed free access to water and food.

**Human donor tissue**. Our research and the protocol for collection of human retinas were conducted in accordance with the tenets of the Declaration of Helsinki and approved by University of Tennessee Health Science Center Institutional Review Board. Informed consent was received from donors or donor family members. Paraffin sections from a human donor eye (70-year-old, male) with no diagnosis of ocular disease were obtained from National Disease Research Interchange (Philadelphia, PA). No donor details apart from age, sex, and cause of death were provided.

**IOP measurement**. IOP of mice was measured using an induction-impact tonometer (Tonolab tonometer, Colonial Medical Supply, Franconia, NH) for rodents according to the manufacturer's recommended procedures. The IOP of rabbits was measured using a Tono-Pen AVIA (Reichert Technologies, Depew, NY) while rabbits were held in a rabbit restrainer. Six consecutive IOP readings were averaged. For each mouse, the tested ophthalmic formulation (10 µl) was applied to the right eye, while the left eye received the blank and serve as a control[29, 60]. One hundred microliters of 0.9% pregabalin eye drops were applied into the inferior conjunctival sac of the right eye of the Dutch belted rabbits, while the left eye received the blank eye drops[61]. All results were expressed as the mean percentage reduction in IOP from baseline (mean% reduction ± SEM).

**Identification of IOP-modulating loci in mice**. IOP data generated from 9.1–13-month-old BXD strains was integrated into the GeneNetwork database. The identification and mapping of phenotypic QTL was performed by linking trait data to genotypes at known genetic marker loci using the WebQTL module on GeneNetwork (http://www.genenetwork.org/)[26, 29, 30, 32].

**Evaluation of *CACNA2D1* in GWAS repositories**. SNPs located within the CACNA2D1 human genomic region were evaluated in the NEIGHBORHOOD consortium metadata[16]. To capture regulatory regions, the genomic locus was defined as all coding sequences + 50 base pairs on either side of the first/last exon.

**eQTL analysis of IOP candidate genes**. IOP data from BXD mice aged 1–2.1 months (64 strains) and 9.1–13 months (65 strains) are publically available as BXD published phenotypes record IDs 16337 (http://www.gn2.genenetwork.org/show_trait?trait_id=16337&dataset=BXDPublish) and 16340 (http://www.gn2.genenetwork.org/show_trait?trait_id=16340&dataset=BXDPublish), respectively. Candidate genes were identified using the tools available on the GeneNetwork/UCSC Genome browser. Gene expression was used as a microtrait to map regulatory eQTLs for the differences in mRNA expression levels over the panel of BXD lines because it is a heritable trait. Whole-eye transcript data from BXD strains available on GeneNetwork as Eye M430v2 (Sep08) RMA (http://www.genenetwork.org/webqtl/main.py?FormID=sharinginfo&GN_AccessionId=207) were used to identify *cis* (locally)-regulated genes from within intervals of interest[34]. The same transcript data were used to identify genes whose expression correlated with IOP at 9.1–13 months of age. Only probes that did not bind to regions containing SNPs were used to avoid hybridization artifacts that may cause differences in expression due to technical error rather than biological variance. All probes were verified by BLAST-like alignment tool (BLAT) (University of California Santa Cruz (UCSC) Genome Browser). Correlation analyses for initial candidate gene selection used Pearson correlation coefficients. Genes were considered as being *cis*-regulated if the associated marker was localized within a 10 Mb distance of the gene position[34].

**Gene set enrichment analysis**. GSEA was performed as described previously using version 2.2.0 (http://www.broadinstitute.org/gsea[35, 62]). Briefly, all nominally significant correlates (uncorrected $P \leq 0.05$, 1603 genes) of *Cacna2d1* were extracted from the whole-eye transcript data and pre-ranked based on their correlation coefficients (Supplementary Data 1). Molecular Signatures Database (MSigDB) version 5.0 was used as the database to perform GSEA. GSEA was performed on these pre-ranked genes using 1000 permutations. This was followed by building modules of related pathways based on at least 25% gene overlap between pathways using the enrichment map strategy. The enrichment score (ES) was calculated as cumulative score from 0. The ES was normalized to account for the size of the gene set being used. To control the expected proportions of false positives, the FDR for *P*-values was calculated using the Benjamini and Hochberg method implemented in LIMMA.

**Immunohistochemistry**. Our standard methods were used to immunolocalize CACNA2D1 in mouse and human eye sections[30, 63, 64]. An anti-CACNA2D1 antibody (Bioss Antibodies, Woburn, MA; catalog #bs-11981R; 1:100) was used in these studies along with goat anti-rabbit Alexa fluor 488 secondary antibody (Invitrogen, Waltham, MA; catalog #A11034; 1:200), and TO-PRO-3 iodide (Invitrogen, Waltham, MA; catalog #T3605) as a nuclear counterstain. Sections were viewed and images were obtained using a Nikon C1 confocal microscope (Nikon, NY).

**Topical pregabalin ophthalmic formulations**. Blank ophthalmic eye drops were prepared by dissolving 2% hydroxypropylmethylcellulose (Sigma-Aldrich, St. Louis, MO) in PBS pH 7.4 to use it as a vehicle (80%) in combination with propylene glycol (10%; Sigma-Aldrich) and PEG 300 (10%; Sigma-Aldrich). Pregabalin (Sigma-Aldrich, St. Louis, MO) ophthalmic eye drops were prepared

using four different concentrations of the drug (0.3, 0.6, 0.9, and 1.2%) by dissolving pregabalin in the previously prepared blank ophthalmic eye drops. These studies were conducted using a single dose–response design. Ten microliters of the pregabalin formulations or the blank ophthalmic eye drops were used in the mouse studies[60] ($n = 4$–6). One hundred microliters of either formulation were used in the rabbit study ($n = 5$). IOP was measured prior to treatment and hourly until the IOP returned to baseline levels.

Several pharmacodynamics parameters were used to evaluate pregabalin including maximum reduction in IOP (% reduction in IOP), time required to reach maximum decrease of IOP ($T_{max}$), time required for IOP to return to baseline (i.e., end of drug effect; $T_{end}$), and total area under the percent reduction in IOP-vs-time curve ($AUC_{total}$). Values of $AUC_{total}$ were determined using a linear trapezoidal method[65]. Results were reported as mean ± SEM.

**Statistics**. To determine how much of the variation in candidate gene across the cohort was due to genetic effects, we calculated the heritability of the candidate gene using the formula of Hegmann and Possidente[26, 32, 33]. Pearson product-moment correlations were calculated using the Correlation matrix tool available in GeneNetwork. Pearson correlation, denoted by $r$, measures the strength of a linear association between two variables, which in our case are IOP (mmHg) and *Cacna2d1* mRNA expression. Partial correlation within-strain and between-strain variances, genotypes vs. IOP, and genotype vs. gene expression were calculated with analysis of variance (ANOVA) using SAS 6.0. (Cary, NC) or GraphPad Prism-7 software (La Jolla, CA). Statistical analyses of the pharmacodynamics results for pregabalin were performed using one-way ANOVA with Tukey–Kramer multiple comparisons test. All pharmacodynamic parameters as well as the statistical analysis of the results were also calculated using GraphPad Prism-7 software (La Jolla, CA).

**Data availability**. The data sets generated during and/or analyzed during the current study are available in the GeneNetwork repository as follows: IOP data from BXD mice aged 1–2.1 months and 9.1–13 months are publically available as BXD published phenotypes record IDs 16337 (http://www.gn2.genenetwork.org/show_trait?trait_id=16337&dataset=BXDPublish) and 16340, (http://www.gn2.genenetwork.org/show_trait?trait_id=16340&dataset=BXDPublish), respectively. *Cacna2d1* expression data is available as Trait ID 1446324_at and 1449999_a_at in the Eye M430v2 (Sep08) RMA database (http://www.genenetwork.org/webqtl/main.py?FormID=sharinginfo&GN_AccessionId=207).

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

## Acknowledgements

We thank Dr. E. Geisert, Dr. L. Lu, and Mr. B. Orr for their assistance in generating the BXD microarray data sets that were used in these analyses. We also thank Dr. H. Lu for acquiring the IOP data sets and Dr. S. Surbhi for assistance with statistical analyses. We further thank S. Ganguli and A. Shepherd for discussion and technical assistance. We acknowledge the financial support from: the Center for Integrative and Translational Genomics at the University of Tennessee Health Science Center; NEI Grants R01EY021200, R01EY022305, and P30EY013080; NIAAA Grant U01AA01666; and a Stein Innovation Award and an unrestricted grant from Research to Prevent Blindness, Inc.

## Author contributions

S.R.C. and M.M.J. conceived the experiments. S.R.C., D.M., and X.W. conducted experiments. S.R.C., D.M., X.W., P.G.H., J.N.C.B., R.W.W., J.L.W., and M.M.J. participated in data interpretation and discussion. M.M.J. conceptualized the project and supervised the experiments. R.W.W. and M.M.J. provided resources/materials/analysis tools for completion of the study. S.R.C. and M.M.J. wrote the manuscript. All authors reviewed and contributed intellectually to the article.

## Additional information

**Competing interests:** The authors declare no competing financial interests.

# NEIGHBORHOOD consortium

Rand Allingham[9], Murray Brilliant[10], Don Budenz[11], John Fingert[12], Douglas Gaasterland[13], Teresa Gaasterland[13], Jonathan L. Haines[4], Lisa Hark[14], Michael Hauser[9], Rob Igo[4], Jae Hee Kang[15], Peter Kraft[16], Richard Lee[17], Paul Lichter[18], Yutao Liu[19], Syoko Moroi[18], Louis R. Pasquale[6,15], Margaret Pericak-Vance[20], Anthony Realini[21], Doug Rhee[22], Julia R. Richards[18], Robert Ritch[23], Joel Schuman[24], William K. Scott[25], Kuldev Singh[26], Arthur Sit[27], Douglas Vollrath[28], Gadi Wollstein[24] & Don Zack[29]

[9]Department of Ophthalmology, Duke University Eye Center, Durham, NC 27705, USA. [10]University of Wisconsin Institute for Clinical and Translational Research, Madison, WI 53705, USA. [11]Department of Ophthalmology, UNC School of Medicine, Chapel Hill, NC 27517, USA. [12]Department of Ophthalmology and Visual Sciences, University of Iowa, Iowa City, IA 52242, USA. [13]Eye Doctors of Washington DC, Washington,

DC 20036, USA. [14]Glaucoma Research Center, Wills Eye Hospital, Philadelphia, PA 19107, USA. [15]Channing Division of Network Medicine, Brigham and Women's Hospital, Boston, MA 02115, USA. [16]Harvard T.H. Chan School of Public Health, Boston, MA 02115, USA. [17]Department of Ophthalmology, Bascom Palmer Eye Institute, University of Miami, Miami, FL 33136, USA. [18]Department of Ophthalmology, Kellogg Eye Center, University of Michigan, Ann Arbor, MI 48105, USA. [19]Department of Cellular Biology and Anatomy, Augusta University, Augusta, GA 30901, USA. [20]John P. Hussman Institute for Human Genomics, University of Miami, Miami, FL 33136, USA. [21]Department of Ophthalmology, West Virginia University, Morgantown, WV 26506, USA. [22]Department of Ophthalmology and Visual Sciences, Case Western Reserve University, Cleveland, OH 44106, USA. [23]Department of Ophthalmology, Einhorn Clinical Research Center, New York Eye and Ear Infirmary, New York, NY 10003, USA. [24]Department of Ophthalmology, NYU Langone Medical Center, New York, NY 10016, USA. [25]Dr. John T. Macdonald Foundation Department of Human Genetics, John P. Hussman Institute for Human Genomics, University of Miami Health System, Miami, FL 33136, USA. [26]Department of Ophthalmology, Stanford University Medical Center, Stanford, CA 94303, USA. [27]Department of Ophthalmology, Mayo Clinic, Rochester, MN 85259, USA. [28]Department of Genetics, Stanford University Medical Center, Stanford, CA 94305, USA. [29]Department of Ophthalmology, Wilmer Eye Institute, The Johns Hopkins University School of Medicine, Baltimore, MD 21287, USA

## International Glaucoma Genetics consortium

Tin Aung[30], Peter Bonnemaijer[31], Cheng-Yu Cheng[31], Jamie Craig[32], Cornelia van Duijn[33], Puya Gharahkhani[34], Adriana Iglesias Gonzalez[35], Christopher J. Hammond[36], Alex Hewitt[37], Rene Hoehn[38], Fridbert Jonansson[39], Anthony Khawaja[40], Chiea Chuen Khor[41], Caroline C.W. Klaver[42], Andrew Lotery[43], David Mackey[44], Stuart MacGregor[45], Calvin Pang[46], Francesca Pasutto[47], Kári Stefansson[48], Gudmar Thorleifsson[48], Unnar Thorsteinsdottir[48], Veronique Vitart[49], Eranga Vithana[30], Terri Young[50] & Tanja Zeller[51]

[30]Singapore Eye Research Institute, National University of Singapore, Singapore, 168751, Singapore. [31]Department of Epidemiology, Erasmus Medical Center, Rotterdam, GE 3015, The Netherlands. [32]Department of Ophthalmology, Flinders University, Adelaide, SA 5042, Australia. [33]Department of Epidemiology, Erasmus Medical Center, Rotterdam, GE 3015, The Netherlands. [34]Statistical Genetics, QIMR Berghofer Medical Research Institute, Brisbane, QLD 4006, Australia. [35]Department of Epidemiology, University Medical Center of Rotterdam, Rotterdam, GE 3015, The Netherlands. [36]Department of Ophthalmology, King's College London, London, SE5 9RS, UK. [37]Department of Ophthlmology, Menzies Institute for Medical Research, University of Tasmania, Tasmania, 7000, Australia. [38]Ophthalmology, Inselspital, University Hospital Bern, University of Bern, Bern, 3010, Switzerland. [39]Faculty of Medicine, University of Iceland, Reykjavik, 101, Iceland. [40]Department of Public Health and Primary Care, University of Cambridge, Cambridge, CB2 1TN, England. [41]Department of Biochemistry, National University of Singapore, Singapore, 117596, Singapore. [42]Department of Ophthalmology, Erasmus Medical Center, Rotterdam, GE 3015, The Netherlands. [43]Clinical and Experimental Sciences, Faculty of Medicine, University of Southampton, Southampton, SO17 1BJ, UK. [44]Centre for Eye Research Australia, University of Melbourne, Royal Victorian Eye and Ear Hospital, Melbourne, VIC 3002, Australia. [45]Statistical Genetics, QIMR Berghofer Medical Research Institute Royal Brisbane Hospital, Brisbane, QLD 4006, Australia. [46]Department of Ophthalmology and Visual Sciences, The Chinese University of Hong Kong, Hong Kong, Hong Kong. [47]Institute of Human Genetics, Friedrich-Alexander-Universität Erlangen-Nürnberg (FAU), Erlangen, 91054, Germany. [48]deCODE genetics/Amgen, Inc., Reykjavik, IS-101, Iceland. [49]MRC Human Genetics Unit, The University of Edinburgh, Edinburgh, EH4 2XU, Scotland. [50]McPherson Eye Research Institute, University of Wisconsin-Madison, Madison, WI 53705, USA. [51]Clinic for General and Interventional Cardiology, University Heart Center Hamburg, Hamburg, 20246, Germany

