## [Transparent Peer Review · Nature Communications]

Reviewer #1 (Remarks to the Author):

The authors present compelling genetic evidence in the mouse showing strong locus control of intraocular pressure (IOP) by a gene locus mapping to murine chromosome 5.

Follow-up criterion-based filtering of candidate genes in the region using i) statistical confidence bounding, ii) positive biological expression with the presence of correlation with IOP, and iii) other downstream, justified selection enabled the authors to focus on *Cacna2d1* as a high priority target. The manuscript is transparently presented, well written and logical throughout.

I have the following suggestions/queries for the authors to consider:

1. Short of an unbiased high content screening approach, it is not possible to definitively pin down *Cacna2d1* as the only gene exerting the control over IOP. Could the authors at least present and discuss (perhaps as a supplement if space does not allow for it in the main text) the other possible targets at the chromosome 5 locus, and why they were finally excluded using the 8-point criteria? How many genes scored 7 out of 8, and were excluded? How many scored 6? A table showing all the candidate genes in the locus and their scores based on the filtering criterion used by the authors would be extremely helpful to the reader
2. The data showing that at baseline, strains with the D allele already have lower IOP compared to strains with the B allele of *Cacna2d1*, were very compelling. When stratified for IOP, strains with the B allele showed higher expression of *Cacna2d1* and significantly higher IOP compared to the D allele, uniformly. This was also very compelling. This reviewer is slightly frustrated that the labelling of the figures was not clear (the Figure legends and Figures themselves do not match), and suggest that the authors clarify this in a revised manuscript.
3. The treatment response, stratified by strain and dosage, was consistent with all the genetic data. It follows that mice with the low IOP D allele would not respond as well as the mice with the higher risk B allele. Overall, the mouse data were compelling and stand alone strongly by itself. The NEIGHBORHOOD consortium data show encouraging, suggestive association at the same locus in humans, but the authors did not clarify which allele is the IOP – increasing / lowering one.
4. Also, there is now a published human IOP GWAS (Hysi et al., Nature Genetics 2014). How does the data square with human IOP loci?

Other questions:

1. What was the baseline IOP in the animal experiments in which pregabalin was administered? I can only find the percentage change in IOP.
2. Do the authors have any conflicts to declare regarding possible patent/IP related to this data?

Reviewer #2 (Remarks to the Author):

Using recombinant inbred mouse system, authors figured out a mouse gene, *Cacna2d1* that regulate IOP. This is an extremely important achievement like *nu* gene identification in case of nude-mouse and *Aly* gene identification in a model mouse, which totally lack lymph node. These achievements are solid and bring important knowledge for the human being. All results using BXD family in this paper are solid and all the statistics were fairly done. Whereas, pharmacological data of pregabalin using only one rabbit (Fig. 5 C) and human GWAS data showing marginal p value are not solid. But, as a starting point for following the high value data obtained by using BXD family, such suggestive data bear reasonable value at this moment.

Minor concerns;

1. They seem to be mistakes; D allele → B allele (p9, line 201) , B allele → D allele (p9, line 202) .
2. P11, line 249: Authors described (refs) and seem to forget to describe the reference numbers which link with names, volumes, pages, titles and authors of some actual papers.
3. Figure 6 was not explained in the main text. " (Fig. 6) " should be added at the end of the sentence "Based upon these collective data~" (p12, line 262) .
4. As for "ns" (not significant) in tables 2 and 3, actual p values should be shown.

We thank the reviewers for their valuable time and effort in reviewing the previous submission of our manuscript. Based upon the very thorough and positive review we received, we have modified the text, tables and figures accordingly. Below, we address each point raised by the reviewers:

Reviewer 1

Query: Short of an unbiased high content screening approach, it is not possible to definitively pin down *Cacna2d1* as the only gene exerting the control over IOP. Could the authors at least present and discuss (perhaps as a supplement if space does not allow for it in the main text) the other possible targets at the chromosome 5 locus, and why they were finally excluded using the 8-point criteria? How many genes scored 7 out of 8, and were excluded? How many scored 6? A table showing all the candidate genes in the locus and their scores based on the filtering criterion used by the authors would be extremely helpful to the reader.

Reply: A revised supplemental table 1 and new supplemental table 2 have been added to address these specific comments.

Query: The data showing that at baseline, strains with the D allele already have lower IOP compared to strains with the B allele of *Cacna2d1*, were very compelling. When stratified for IOP, strains with the B allele showed higher expression of *Cacna2d1* and significantly higher IOP compared to the D allele, uniformly. This was also very compelling. This reviewer is slightly frustrated that the labelling of the figures was not clear (the Figure legends and Figures themselves do not match), and suggest that the authors clarify this in a revised manuscript.

Reply: I apologize for any errors in labeling the figures. All figures and legends have been revised so that they correct labeling is in the current revised version of the manuscript.

Query: The treatment response, stratified by strain and dosage, was consistent with all the genetic data. It follows that mice with the low IOP D allele would not respond as well as the mice with the higher risk B allele. Overall, the mouse data were compelling and stand alone strongly by itself. The NEIGHBORHOOD consortium data show encouraging, suggestive association at the same locus in humans, but the authors did not clarify which allele is the IOP – increasing / lowering one.

Reply: A new supplemental table 5 was created to expand on the human POAG GWAS data and ~~also to~~ include the human IOP GWAS data. In all cases, allele 1 is the effect allele.

Query: Also, there is now a published human IOP GWAS (Hysi et al., Nature Genetics 2014). ~~How~~ does the data square with human IOP loci?

Reply: The data of Hysi et al are now included in the manuscript on page 8 and supplemental table 5.

Query: What was the baseline IOP in the animal experiments in which pregabalin was administered? I can only find the percentage change in IOP.

Reply: Baseline IOPs have been added to all tables.

Query: Do the authors have any conflicts to declare regarding possible patent/IP related to this data?

Reply: No, we have no conflicts to declare.

Reviewer 2

Query: Using recombinant inbred mouse system, authors figured out a mouse gene, Cacna2d1 that regulate IOP. This is an extremely important achievement like nu gene identification in case of nude-mouse and Aly gene identification in a model mouse, which totally lack lymph node. These achievements are solid and bring important knowledge for the human being. All results using BXD family in this paper are solid and all the statistics were fairly done. Whereas, pharmacological data of pregabalin using only one rabbit (Fig. 5 C) and human GWAS data showing marginal p value are not solid. But, as a starting point for following the high value data obtained by using BXD family, such suggestive data bear reasonable value at this moment.

Reply: We have evaluated the efficacy of pregabalin in five DB rabbits and include that data on page 10 of the manuscript and table 3.

Query: 1. They seem to be mistakes; D allele → B allele (p9, line 201) , B allele → D allele (p9, line 202).

Reply: I apologize for any errors in the manuscript. The paper has been revised throughout to correct any labeling errors.

Query: P11, line 249: Authors described (refs) and seem to forget to describe the reference numbers which link with names, volumes, pages, titles and authors of some actual papers.

Reply: This has been corrected.

Query: Figure 6 was not explained in the main text. “ (Fig. 6) ” should be added at the end of the sentence “Based upon these collective data~” (p12, line 262).

Reply: Figure 6 is now cited on page 12.

Query: As for “ns” (not significant) in tables 2 and 3, actual p values should be shown.

Reply: Actual p values are now presented in tables 1 and 2.

Reviewer #1 (Remarks to the Author):

The authors have satisfactorily responded to my comments and have revised the manuscript well.

Reviewer #2 (Remarks to the Author):

The study by Chintalapudi et al. is the first bidirectional study that has successfully identified an IOP-modulating candidate gene in glaucoma and demonstrated a differential response to therapy in mouse. Association with human GWAS was shown although neither p value nor odds ratio are strong value.

Authors corrected all the queries pointed out by reviewers.